# Chiral Recognition R- and RS- of New Antifungal: Complexation/Solubilization/Dissolution Thermodynamics and Permeability Assay

**DOI:** 10.3390/pharmaceutics14040864

**Published:** 2022-04-15

**Authors:** Tatyana V. Volkova, Olga R. Simonova, Igor B. Levshin, German L. Perlovich

**Affiliations:** 1G.A. Krestov Institute of Solution Chemistry RAS, 153045 Ivanovo, Russia; vtv@isc-ras.ru (T.V.V.); ors@isc-ras.ru (O.R.S.); 2Gause Institute of New Antibiotics, 119021 Moscow, Russia; levshiniv20001@rambler.ru

**Keywords:** chiral discrimination, complexation, solubilization, dissolution, enantioselective transport

## Abstract

Novel potential antifungal of 1,2,4-triazole class have been synthesized as pure enantiomer (R-98) and racemic (RS-186). The effect of 2-hydroxypropyl-β-cyclodextrin (*CD*) on the solubility and permeability of RS-186 and R-98 in terms of chiral recognition was investigated. Phase solubility studies were carried out at 4 temperatures in 0–0.05 M *CD* concentration range for pH 2.0 and pH 7.4. $A_{L}$- and $A_{L}^{-}$-type phase-solubility profiles were obtained for both compounds in pH 2.0 and pH 7.4. The racemic formed more stable complexes with *CD* as compared to R-isomer. Disclosing of chiral discrimination was facilitated using the approach based on the complex consideration of the derived complexation/solubilization/inherent dissolution thermodynamic functions, including the differential parameters between the racemic compound and R-enantiomer. The differences in the thermodynamic parameters determined by the chirality were discussed in terms of the driving forces of the processes and the main interactions of the compounds with *CD* in solution. The membrane permeability of both samples in the presence of *CD* was accessed in order to evaluate the specificity of enantioselective transport through the lipophilic membrane. The solubility/permeability interrelation was disclosed. The investigated compounds were classified as medium permeable in pure buffers and low permeable in the presence of 0.01 M *CD*. The obtained results can be useful for the design of pharmaceutical products in the form of liquid formulations based on the investigated substances.

## 1. Introduction

Physicochemical properties of racemic and pure enantiomer of pharmaceutically active compounds can often vary, thus resulting in differences in their biological activities because chirality is a crucial factor for the receptor-binding interactions [1]. Moreover, one of the isomers can appear to be not only ineffective but even toxic [2] and should be separated and withdrawn from further drug design process. Evaluation of the stereospecificity of enantiomers is highly demanded in quality control of pure drugs in the light of timely discarding of undesirable samples and registering the interconversion of enantiomers [3]. It is logical to assume that stereoselectivity can become apparent in the transport through the biomembranes and cellular uptake, if the enantiomers pass them in different ways, thus influencing the pharmacokinetic and pharmacodynamic properties. The literature resources disclose the situations when different permeability for racemic against pure enantiomers was observed for a number of drugs [4]. At the same time, the authors [5] revealed some compounds (atenolol, sotalol, metoprolol and propafenone) for which the permeability is not enantioselective. Regardless, it is reasonable to consider the pharmacodynamic/pharmacokinetic profile of each chiral drug as a racemic mixture and individual stereoisomers in order to realize the rationality of using racemic/enantiomer for the most beneficial result.

A number of methods aimed at the separation of the racemic mixtures have been developed, including the mechanical, biochemical, chemical, and electrochemical. The most applied are capillary electrophoresis, nuclear magnetic resonance spectroscopy, chromatography [6,7,8]. In these methods, mainly, macrocyclic compounds are used as chiral selectors. Among them, cyclodextrins (*CD*) are often applied due to the presence of a non-polar cavity which can include, completely or partly, the organic molecules thus forming the inclusion complexes [9]. The binding mode of the interactions between the guest molecules and *CD*s depends on the dimensions of the organic compound and *CD* cavity, as well as their nature, ionization state, etc. The advantage of *CD*s (oligomers of α-D-glucopyranose) as chiral selectors is based on their ability to form diastereomeric inclusion complexes with enantiomeric pairs due to the intrinsic chirality [10,11].

Importantly, in order to disclose the nature of the specific *CD* enantioselectivity towards a certain pair of enantiomers in solution, a thermodynamic study from the solubility at different temperatures can be very useful. Thus, Rekharsky and Inoue [12], based on a great number of enantiomers, recognized the interactions and properties crucial for precise chiral separation. Moreover, the authors suggested that the knowledge on the thermodynamic parameters allows us to gain better insight into the stereochemistry and functional groups of the guest, as well as the mode of inclusion complexation.

The objects of our investigation were the racemic compound (RS-186) and pure enantiomer (R-98), which were synthesized as a potential antifungal of 1,2,4-triazole class (Figure 1).

Since the causative agents of fungal infections will not become a thing of the past, but rather mutate and acquire new modes of action, a need for effective new antifungals remains constant. Expectedly, an issue of enantiorecognition also arises for drugs against various fungi. For example, Li et al. [13] disclosed a role of difference in binding energy in the enantioseparation process by the example of complex formation of 12 new triadimenol and seven new triazole antifungal active compounds with heptakis-(2,3,6-tri-O-methyl)-β-cyclodextrin. In their turn, Wearley et al. [14] estimated the relationship between physicochemical properties (melting point, solubility), including skin permeation of the racemic and pure enantiomers of topical antifungal of 1,2,4-triazole class, and found no linear correlation between the efficacy of topically active drugs and the thermodynamic activity in the vehicle or the skin.

In the present investigation, we revealed the effect of 2-HP-β-cyclodextrin (*CD*) on the solubility of the racemic compound (RS-186) and pure enantiomer (R-98). The solubility results were used for determination and comparative analysis of the complexation/solubilization/inherent dissolution thermodynamic functions with the aim of disclosing the driving forces and the main interactions governing the processes in solutions and thermodynamic bases of the chiral recognition by *CD*. Two aqueous buffers—pH 2.0 and pH 7.4—were applied in order to reveal the influence of the pH on the outlined parameters. In addition, membrane permeability of both samples in the presence of *CD* was accessed in two buffer solutions in order to evaluate the specificity of enantioselective transport through the lipophilic membrane-PermeaPad barrier. We hope that the obtained results would be beneficial for the design of liquid formulations based on the investigated substance and for the evaluation of the behavior in the physiological fluids of different acidities.

## 2. Materials and Methods

### 2.1. Materials

New racemic compound (RS-186) and pure enantiomer (R-98) of 1,2,4-triazole class derivative (Figure 1) were synthesized in Gause Institute of New Antibiotics (Moscow, Russia).

β-Cyclodextrin (CAS 7585-39-9, M = 1134.98 g·mol^−1^ (0.6 molar substitution), purity ≥ 97.0%), 2-hydroxypropyl-β-cyclodextrin (CAS 128446-35-5, M~1380 g·mol^−1^, purity ≥ 96.0%) were obtained from Sigma-Aldrich (St. Louis, MO, USA).

PermeaPad barrier was obtained from Labtastic distributor (PHABIOC, Germany, Espelkamp, www.permeapad.com, accessed on 20 May 2021).

Potassium dihydrogen phosphate (purity ≥ 99%) and disodium hydrogen phosphate dodecahydrate (purity ≥ 99%) were obtained from Merk (Darmstadt, Germany); potassium chloride (purity ≥ 99%) and hydrochloric acid 0.1 mol·dm^−3^ fixanal were obtained from Aldrich.

Buffer preparation: phosphate buffer pH 7.4 (I = 0.15 mol·L^−1^) was prepared by combining KHPO_4_ (9.1 g in 1 L) and NaH_2_PO_4_·12H_2_O (23.6 g in 1 L). In order to obtain buffer solution pH 2.0 (I = 0.10 mol·L^−1^) 6.57 g KCl was dissolved in water, and 119.0 mL of 0.1 mol mol·L^−1^ hydrochloric acid were added. The volume of the solution was adjusted to 1 L with water. A FG2-Kit pH meter (Mettler Toledo, Greifenzee, Switzerland) standardized with pH 1.68, 6.86 and 9.22 solutions was used to measure the pHs of the solutions.

Bidistilled water (electrical conductivity equal to 2.1 μS cm^−1^) was used for the preparation of buffer solutions. The solvents and reagents were used as received.

### 2.2. Methods

#### 2.2.1. Synthesis

Synthetic procedure was applied to the synthesis of racemic compound (RS-186) and pure enantiomer (R-98):

(Z)-5-(4-chlorobenzylidene)-3-(3-(4-(2-(2,4-difluorophenyl)-2-hydroxy-3-(1H-1,2,4-triazol-1-yl)propyl)piperazin-1-yl)-2-hydroxypropyl)thiazolidine-2,4-dione is presented in Appendix A, respectively. The step-by-step description of the synthesis process is given just after Appendix A. The respective ^1^H NMR and ^13^C NMR spectra were recorded as follows:

RS-186.

^1^H NMR (600 MHz, DMSO-d6) δ 8.26 (s, 1H), 7.89 (s, 1H), 7.71 (s, 1H), 7.67–7.56 (m, 4H), 7.36 (dq, J = 16.6, 8.4 Hz, 1H), 7.15–7.06 (m, 1H), 6.92 (t, J = 8.2 Hz, 1H), 5.03 (s, 1H), 4.51 (s, 2H), 3.90 (t, J = 6.6 Hz, 1H), 3.68–3.58 (m, 2H), 3.55–3.32 (m, 1H), 3.28 (t, J = 7.0 Hz, 3H), 2.68 (s, 4H), 2.53–2.46 (m, 2H), 2.32 (d, J = 19.1 Hz, 6H), 2.16 (t, J = 8.1 Hz, 3H), 1.89 (q, J = 7.4 Hz, 4H).

^13^C NMR (151 MHz, DMSO-d6) δ: δ 167.37, 165.73, 163.24, 163.15, 161.60, 161.51, 159.85, 159.78, 158.22, 149.26, 144.83, 135.21, 131.94, 131.71, 131.23, 130.06, 129.53, 123.29, 122.45, 111.52, 111.39, 104.65, 104.48, 104.31, 72.90, 62.58, 62.57, 62.03, 55.52, 48.59, 48.53, 45.27, 29.04.

R-98.

^1^H NMR (600 MHz, DMSO-d6) δ 11.16 (s, 1H), 8.77 (s, 1H), 8.00 (s, 1H), 7.90 (s, 1H), 7.70–7.63 (m, 2H), 7.63–7.56 (m, 2H), 7.47–7.38 (m, 1H), 7.24 (ddd, J = 11.8, 9.0, 2.6 Hz, 1H), 7.01 (t, J = 8.5 Hz, 1H), 4.85 (d, J = 14.4 Hz, 1H), 4.72 (dd, J = 14.4, 3.6 Hz, 1H), 4.32 (d, J = 8.7 Hz, 1H), 3.75 (hept, J = 6.1 Hz, 1H), 3.68 (dd, J = 13.7, 7.3 Hz, 1H), 3.62–3.55 (m, 1H), 3.51 (s, 10H), 3.35–3.25 (m, 1H).

The graphical ^1^H NMR and ^13^C NMR spectra for the synthesized compounds with the corresponding peak integrals are presented in the Appendix A. The purity of R-186 and R-98, subjected for further investigations, was 99.5%.

#### 2.2.2. Powder X-ray Diffraction (PXRD)

The powder XRD data were recorded under ambient conditions on a D2 Phaser (Bragg-Brentano) diffractometer (Bruker AXS, Karlsruhe, Germany) with a copper X-ray source (λCuKα1 = 1.5406 Å) and a high-resolution position-sensitive LYNXEYE XE T detector. The samples were placed into the plate sample holders and rotated at a speed of 15 rpm during the data acquisition.

#### 2.2.3. UV-Vis Spectrophotometry

UV-vis spectra of the individual compound in pure buffers, n-hexane, as well as inclusion complexes with *CD*, were obtained with Cary 50 spectrophotometer (Varian, Palo Alto, CA, USA, Software Version 3.00 (339)) using a 1 cm quartz cell. The scans were registered from 200 to 500 nm.

#### 2.2.4. Phase Solubility Studies

Phase solubility studies were performed at four temperature points: (298.15–313.15) K using the standard shake-flask method in two aqueous buffered solutions at pH 2.0 and pH 7.4. In addition, four concentrations of 2-hydroxypropyl-β-cyclodextrin 0.01; 0.03; 0.04; 0.05 M were used to construct the phase solubility diagrams. An excess amount of the substance was placed in the glass screw-capped vials with the solvent. The vials were stirred in an air thermostat for a certain period of time (a period of 3 days was estimated to be enough for reaching the equilibrium between the solid and the solvent). Then, the saturated solutions were kept in a thermostat in mild conditions no less than 12 h and thereafter centrifuged at a fixed temperature (Biofuge pico, Thermo Electron LED GmbH, Langenselbold, Germany) for 10 min at 12,000 rpm. If necessary, the saturated solution was diluted with the corresponding solvent to the concentration adequate for spectrophotometric assay (Cary-50 Varian, Palo Alto, CA, USA, Software Version 3.00 (339)). Both the racemic and enantiomer showed the same spectrum in the UV range. The absorbance of the solutions was used for the equilibrium solubility determination with the help of calibration curves at each pH (2–4% accuracy). The experimental results were reported as an average of at least three replicated experiments. Temperature uncertainty was ±0.1 K. The sensitivity of UV spectroscopic determinations was characterized by the values of the limit of detection (LOD)/limit of quantification (LOQ). LOD/LOQ = 2.61/7.92 μM and LOD/LOQ = 3.56/10.80 μM for R-98 and RS-186, respectively (buffer pH 2.0). LOD/LOQ = 0.40/1.22 μM and LOD/LOQ = 0.30/0.92 μM for R-98 and RS-186, respectively (buffer pH 7.4).

#### 2.2.5. Complexation Parameters Determination

Thermodynamic functions of the complexation process were determined from the phase solubility diagrams following Higuchi and Connors [15]. The linear *A_L_*-type phase-solubility diagrams evidenced the complexes of the first order in respect to cyclodextrin. The equilibrium of the complex formation with 1:1 stoichiometry in the cyclodextrin solution can be described in the following manner:(1)$$D+CD\overset{K_{C}}{↔}D\cdot CD$$
where *D* is the drug molecule, *CD* is the cyclodextrin molecule, and *D·CD* is drug/cyclodextrin complex. Respectively, the complexation constant ($K_{C}$) can be expressed by the equation:(2)$$K_{C}=\frac{\left[D\cdot CD\right]}{\left[D\right]\left[CD\right]}$$
where [*D·CD*] is the complex concentration, [*CD*] is the cyclodextrin concentration, and [*D*] is the drug concentration. A plot of total solubility ($S_{2}$ expressed in molarity—M) in the presence of *CD* versus *CD* concentration ($C_{CD}$) (a phase-solubility profile) gives a straight line with a slope equal to $\left(K_{C}^{S}\cdot S_{2}^{0}/(1+K_{C}^{S}\cdot S_{2}^{0})\right)$ and intercept equal to $S_{2}^{0}$ from which the apparent stability constants ($K_{C}^{S}$) were estimated via Higuchi–Connors equation [15]:(3)$$K_{C}^{S}=\frac{slope}{S_{2}^{0}\cdot \left(1-slope\right)},$$
where $S_{2}^{0}$ is the intrinsic solubility of the drug at designated pH in the absence of *CD* and the slope is the corresponding slope of the phase-solubility diagram.

The solubilizing potential of cyclodextrin is prominently determined by the complexation efficiency (*CE*) equal to the complex to free cyclodextrin concentration ratio, and was calculated from the slope of the phase-solubility profile by the following equation [16]:(4)$$CE=S_{2}^{0}\cdot K_{C}^{S}\cdot 100\%=\frac{\left[D\cdot CD\right]}{\left[CD\right]}\cdot 100\%=\frac{slope}{\left(1-slope\right)}\cdot 100\%,$$

From *CE*, the (*D*:*CD*) ratio can be derived, which is useful for the estimation of the solubility improvement in *CD* solutions. Using the *CE* value, the molar ratio (*Drug*:*CD*) was determined [16]:(5)$$Drug:CD=1:\left(1+\frac{1}{CE}\right)$$

#### 2.2.6. In Vitro Permeability Assay

Permeability coefficients of RS-186 and R-98 were measured through the PermeaPad barrier (PHABIOC, Germany, Espelkamp, www.permeapad.com, accessed on 20 May 2021) of an effective surface area of 0.785 cm^2^ in a vertical type of Franz diffusion cell (PermeGear, Inc., Hellertown, PA, USA) with 7 mL volume. PermeaPad is a biomimetic membrane proposed by di Cagno and Bauer-Brandl [17] for the fast screening of the permeability of the marketed drugs and drug-like compounds. In brief, the barrier was mounted between the donor and receptor chambers. The donor chamber was filled with a stock solution of the investigated substance in buffer solution at pH 2.0 or pH 7.4 and was stirred vigorously during the permeability experiment. The receptor compartment contained 1 mL of the respective fresh buffer. The system was thermostated at 310.15 ± 0.1 K. A probe of 0.5 mL was withdrawn every 30 min from the receptor chamber and replaced with an equal amount of the respective fresh buffer. The permeation process followed the reverse dialysis [18]. The withdrawn samples were analyzed in 96-well UV black plate (Costar) via spectrophotometer (Spectramax 190; Molecular devices, Molecular Devices Corporation, California, CA, USA) at the appropriate wavelength. The permeation profiles were plotted as the amount of the permeated drug over the surface area (dQ/A) versus the time (t). The flux (J) was calculated as a slope of permeation profiles according to the equation:(6)$$J=\frac{dQ}{A\times dt}$$

The apparent permeability coefficient ($P_{app}$) was calculated by normalizing the flux measured over the concentration of the drug in the donor compartment ($C_{0}$), as described by the equation:(7)$$P_{app}=\frac{J}{C_{o}}$$

Each permeability experiment was repeated at least 3 times and the average value of $P_{app}$ was determined. The sink conditions during the experiment were realized: the drug concentration in the acceptor chamber did not exceed 10% of the drug concentration in the donor chamber at any time.

## 3. Theoretical Section

Before presenting the experimental results that address the main question of the study, it is instructive to provide the main issues on which the applied theoretical approach is based.

Issue 1. Pharmaceutical systems of drug compound containing solubilizing additives (2-HP-β-cyclodextrin in our case) are of great interest from both a practical and a theoretical point of view. Mostly, they are complex multicomponent ones, and their dilution or thermodynamic properties can be adjusted empirically according to the specificity of application. At the same time, model calculations to predict such properties are very useful. Since the thermodynamic properties can be precisely measured, they can serve as a suitable tool for the quantitative evaluation of the processes occurring in multicomponent solutions. In our previous study, on the example of a novel potential antifungal compound [19], we applied an approach aimed at the understanding of the driving forces of the molecule transition from pure buffers to the *CD* solutions upon the simultaneous solubilization and complex formation. Since the magnitudes of the standard free energy and entropy changes depend on the standard state, the mole fraction unitary units which provide better chemical interpretation were used for an accurate comparative analysis of the solubililization and complexation processes in the *CD* solution. Taking into account that the dissolution occurring in the inherent (without *CD*) buffered solutions can influence the equilibrium along with the solubililization and complexation, most probably, the binding equilibrium is determined by the mutual interplay of all the outlined processes. Due to this, in the present study we took the inherent dissolution thermodynamic parameters into consideration.

The Gibbs energy change of the inherent (in pure aqueous buffer without *CD*) dissolution ($\Delta G_{inh/sol}^{0,X}$) in mole fraction scale was derived from the compound mole fraction solubility ($X_{2}^{0}$) in the following way:(8)$$\Delta G_{inh/sol}^{0,X}=-RT\ln X_{2}^{0}$$

The standard solution enthalpies $\Delta H_{inh/sol}^{0,X}$ and entropies $\Delta S_{inh/sol}^{0,X}$ were calculated using a linear dependence between $\ln X_{2}^{0}$ and 1/T and van’t Hoff equation:(9)$$\ln X_{2}^{0}=-\frac{\Delta H_{inh/sol}^{0,X}}{RT}+\frac{\Delta S_{inh/sol}^{0,X}}{R}$$

The free energy of transferring a compound from the aqueous buffer solution to the *CD* solution of a certain *CD* concentration ($C_{CD}$) (corresponds to solubilization process) was calculated by Equation (10):(10)$$\Delta G_{slbz}^{0,X}(C_{CD})=-RT\ln\frac{X_{2}}{X_{2}^{0}},$$
where $X_{2}^{0}$ is the inherent solubility of the drug in mole fraction in the absence of *CD* at temperature *T*, and $X_{2}$ is the total solubility in the presence of *CD* at the same temperature, R is the universal gas constant. The solubilization enthalpy ($\Delta H_{slbz}^{0,X}(C_{CD})$) and entropy ($T\Delta S_{slbz}^{0,X}(C_{CD})$) were derived as follows:(11)$$\Delta H_{slbz}^{0,X}(C_{CD})=\Delta H_{sol}^{0,X}(C_{CD})-\Delta H_{sol}^{0,X}$$
(12)$$T\Delta S_{slbz}^{0,X}(C_{CD})=T\Delta S_{sol}^{0,X}(C_{CD})-T\Delta S_{sol}^{0,X}$$
where $\Delta H_{sol}^{0,X}$ and $\Delta S_{sol}^{0,X}$ are the enthalpy and entropy of solubility process of the investigated compounds in the absence of *CD*, whereas $\Delta H_{sol}^{0,X}(C_{CD})$ and $\Delta S_{sol}^{0,X}(C_{CD})$ are the enthalpy and entropy of solubility process in the presence of *CD* with concentration $C_{CD}$.

In order to determine the thermodynamic parameters of the complex formation for the compounds with *CD*, the stability constants ($K_{C}^{X}$, mole fraction scale) at different temperatures were obtained using the phase diagrams and mole fractions ($X_{2}^{0}$) (analogously to Equation (3) by the following equation:(13)$$K_{C}^{X}=\frac{slope}{X_{2}^{0}\cdot \left(1-slope\right)},$$

Respectively, the Gibbs energy of the complexation in mole fraction scale ($\Delta G_{C}^{0,X}$) was derived from the stability constants ($K_{C}^{X}$) in the following way:(14)$$\Delta G_{C}^{0,X}=-RT\ln K_{C}^{X}$$

The thermodynamic parameters of the complexation, i.e., the standard enthalpy change ($\Delta H_{C}^{0,X}$) and the standard entropy change ($\Delta S_{C}^{0,X}$) were calculated with the help of the integral form of the van’t Hoff’s equation:(15)$$\ln K_{C}^{X}=-\frac{\Delta H_{C}^{0,X}}{RT}+\frac{\Delta S_{C}^{0,X}}{R}$$
where a plot of $\ln K_{C}^{X}$ versus 1/T produces Slope = $-\frac{\Delta H_{C}^{0,X}}{R}$ and intercept equal to $\frac{\Delta S_{C}^{0,X}}{R}$.

In order to understand the contribution of the hydrophobic effect (desolvation) to the complexation process on a relative scale, the approach disclosing the correlation of the free energy of complex formation with the free energy of the substance solubility in the same buffer solution proposed by Al Omari et al. [20] was applied. To this end, the Gibbs energy of complex formation ($\Delta G_{C}^{T,X}$) and inherent dissolution ($\Delta G_{inh/sol}^{T,X}$) in mole fraction scale were used. A linear correlation between $\Delta G_{C}^{T,X}$ and $\Delta G_{sol}^{T,X}$ results in Equation (16):(16)$$\Delta G_{C}^{T,X}=a\cdot \Delta G_{inh/sol}^{T,X}+b$$
where the slope (*a*) is the magnitude of the hydrophobic effect, and the intercept (*b*) is the structural factor.

Issue 2. The ability of *CD*s to form diastereomeric inclusion complexes with enantiomeric pairs due to the intrinsic chirality gives us an opportunity to disclose the nature of the specific *CD* enantioselectivity towards a specific pair of enantiomers in solution. As was shown by Rekharsky and Inoue [12], a thermodynamic study from the solubility of enantiomers and racemate at different temperatures can be very useful in the disclosing of the interactions and properties determining the process of chiral separation and the mode of inclusion complexation. The application of the approach described above (Issue 1) to the enantiomer/racemate discrimination provides a more detailed insight into the enantiomeric differentiation in respect to the two considered processes: complexation and solubilization. Moreover, similar to Yonemochi et al. [21], the free energy difference of the complexation, solubilization and inherent dissolution of the racemic compound and its enantiomer can be calculated using Equations (17)–(19), respectively:(17)$$\Delta \Delta G_{C}^{0,X}(RC-E)=\Delta G_{C}^{0,X}(RC)-\Delta G_{C}^{0,X}(E)$$
(18)$$\Delta \Delta G_{slbz}^{0,X}(RC-E)=\Delta G_{slbz}^{0,X}(RC)-\Delta G_{slbz}^{0,X}(E)$$
(19)$$\Delta \Delta G_{inh/sol}^{0,X}(RC-E)=\Delta G_{inh/sol}^{0,X}(RC)-\Delta G_{inh/sol}^{0,X}(E)$$
where *RC* is the racemic compound and *E* is enantiomer. The differences of the complexation/solubilization/dissolution enthalpy ($\Delta \Delta H^{0,X}(RC-E)$) and entropy ($\Delta \Delta S^{0,X}(RC-E)$) can be derived in the same manner.

Hereby, combining the results obtained using Issues 1 and 2, the racemate/enantiomer differentiation thermodynamics can be evaluated comprehensively.

## 4. Results

### 4.1. UV–Vis Spectroscopy

The UV-vis spectroscopy is a convenient tool for the evaluation of the “host-guest” interactions from a spectral shift of the peak on the absorption curve when the *CD* is added to the solution [22]. The absorption spectra of the studied substances in pure buffers at pH 2.0 and pH 7.4, as well as with additions of 2-HP-β-*CD*, were measured within the range of 200–450 nm and are illustrated in Figure 2. Since there was no difference in the absorption of R-98 and RS-186 was detected, only the spectrum of RS-186 is shown.

A slight difference between the absorption maxima at pH 2.0 (λ = 333 nm and 239.0 nm) and pH 7.4 (λ = 332.1 nm and 235 nm) was observed, obviously, due to an ionized state of the molecules in acidic medium. Weak basic properties of the studied compounds are recognized from the proton affinity of a triazole ring nitrogen atom (as in fluconazole) and both piperazine and triazole ring nitrogens (as in itraconazole) with a predominant protonation of a piperazine. It is likely that a blue shift upon the decrease of the acidity of the buffer was due to an increase in the hydrogen bonding capacity of the solvent [23]. As opposed, no shifting was observed to the structurally analogous compound L-173 (its structure is given in Appendix A) investigated in our previous study [19]. It is well known that the UV spectrum is sensitive to the dielectric properties of the medium [24]. Obviously, the dielectric constant of *CD* is lower than that of pure water, leading to the shifting of the absorption maximum in the cyclodextrin solution. As is shown in Figure 2, addition of *CD* causes only a slightly blue shift (2 nm) of the absorption band, thus indicating a rather weak interaction of the compounds with *CD*. At the same time, a hypsochromic effect in the presence of *CD* indicates the influence of the *CD* hydrophobic cavity (less polar environment as compared to pure buffer) in the complexation process, as was also shown by Ghosh et al. [25] for β*CD* solutions of several tetrahydrocarbazol derivatives. In order to disclose this phenomenon, a spectrum of RS-186 was recorded in n-hexane (Figure 2), non-polar solvent, a model of the hydrophobic environments. As follows from Figure 2, the blue shifts upon the addition of *CD* to the dissolution medium are toward the absorption maximum observed in n-hexane spectrum, once more proving the interaction of the chromophore of the studied compound with the hydrophobic inner cavity of *CD*.

### 4.2. Solubility Studies

Solubility experiments were carried out in two aqueous buffers (pH 2.0 and pH 7.4) in order to reveal the impact of the pH and ionization states of the molecules on the complexation process. Two buffer solutions: acidic (pH 2.0) and pH 7.4 were used since they mimic the biological fluids (gastric fluid and the blood plasma, respectively). Notably, the solubility experiments in β-cyclodextrin solutions were tried but were ultimately unsuccessful. An insoluble complex precipitated, causing turbidity of the solvent and lowering of the compound concentration (results not shown) due to limited β-cyclodextrin aqueous solubility (about 18.5 mg·mL^−1^). Experimental solubility of single enantiomer (R-98) and racemic compound (RS-186) in pure buffer solutions at pH 2.0 and pH 7.4 at HP-β-*CD* concentrations (0.01; 0.03; 0.04; 0.05 M) in the temperature range of 298.15–313.15 K is listed in Table 1. The temperature interval includes 310.15 K, very close to the temperature of the human body and 298.15 K, the standard temperature often used for the evaluation of the thermodynamic parameters.

As follows from the data in Table 1, the solubility reveals pH-dependent behavior for both R-98 and RS-186. In pure buffer solutions it varies from 1.8 × 10^−5^/2.7 × 10^−5^ M at pH 7.4 to 6.2 × 10^−4^/5.0 × 10^−4^ M at pH 2.0 for R-98/RS-186, respectively. Bearing in mind that the investigated structure is very close to the previously reported [19,26] L-173 (Appendix A) it was interesting to compare the solubility of these substances. Besides this, both L-173 and the investigated compounds include a fragment of fluconazole (FCZ) (the structure is shown together with L-173 in Appendix A), thus making it reasonable to compare this well-studied antifungal with the other. Similar to the reported substances, the solubility of the studied ones is higher in the acidic pH as compared to pH 7.4, due obviously to the presence of the ionized particles of the compounds in acidic medium. At the same time, the following order of the solubility is detected: pH 2.0: (FCZ) > (RS-186 ≥ R-98) > (L-173); pH 7.4: (FCZ) > (R-98 ≥ RS-186) >> (L-173). Evidently, a pronounced difference between the solubility of the structurally similar substance L-173 and the studied compounds can be due to the amorphous state [27] of the last ones proved by the PXRD experiments (Appendix A) for R-98 (for RS-186 the PRXD pattern is not shown, since it is identical to R-98). The PRXD patterns represent an aureole, indicating an amorphous disordered structure with the isotropic properties of the solid phases and absence of crystalline regions. According to the literature, there is no strict regularity between the solubility of racemic and enantiomer. In the present study, an opposite result in different pH were obtained. Exactly, at 298.15 K in buffer solution pH 2.0 a 1.24-fold higher solubility value for enantiomer R-98 as compared to RS-186 was determined. In opposition, within pH 7.4, the solubility of RS-186 was 1.5-fold greater than R-98. For the comparison, El-Arini et al. [28] reported the solubility of (+)-praziquantel in distilled water as 1.78-fold higher than for (±)-praziquantel. In their turn, a 3.7-fold greater solubility of racemic compound, as compared to enantiomer, was detected by Yonemochi et al. [21] for propranolol HCl. Interestingly, Nerurkar et al. [29] revealed the solubility of racemic ibuprofen in unbuffered water to be 1.5-fold higher than the respective enantiomer. At the same time, the investigation of racemic/enantiomer ibuprofen in aqueous HCl/KCl buffer solution at pH 1.5 [30] showed at about a twice greater solubility of enantiomer against the racemate at 20 °C and 38 °C. These results are in agreement with our solubility data in different pHs. Since the compounds investigated in the present study are lipophilic and poorly soluble in aqueous buffers, they dissolve mainly via hydrophobic interactions (hydrophobic hydration) and, in addition, the dipole–dipole interactions between the ionized groups and hydrogen bonding sites on the water molecules can also facilitate the solubility. The tendency of water molecules to self-association promotes a squeezing of the bulky aromatic substance molecules from the water structure, thus impeding the dissolution process. At the same time, the addition of cyclodextrin (containing a hydrophobic cavity) in solution promotes the interactions of the lipophilic substances with the *CD* cavity and increases the solubility. Notably, the solubility of both compounds in pure buffers, as well as in an acidic medium (pH 2.0), increases with temperature growth. At the same time, in buffer solution at pH 7.4 in the presence of *CD*, a dramatic solubility reduction is detected. Namely, an 11-fold decrease is observed for both compounds upon the transition from 298.15 K to 313.15 K at maximal *CD* concentration (0.05 M). As follows, the mode of interaction of the compounds with the *CD* molecule is different for neutral and ionized species. A similar rather scarce regularity was reported by, for example, Peeters et al. [31] for alfaxalone complexation with 2-hydroxypropyl-β-cyclodextrin and by Tommasini et al. [32] for flavonoids hesperetin and naringenin in aqueous β-cyclodextrin solution. This regularity seems to be crucial in the case of the formulations based on the investigated compound and *CD*, and requires strict temperature control of the storage. For both racemic and R-enantiomer compounds a significantly more pronounced effect of *CD* on the solubility at 298.15 K was revealed in buffer pH 7.4 (205/148-fold), as compared to acidic buffer (21/28-fold) for R-98/RS-186, respectively. This result follows a general trend: the lower the solubility, the more the solubility enhance ratio [33]. Moreover, a neutral form of both compounds in buffer pH 7.4 exhibits a higher affinity for *CD* than the ionic particles in an acidic medium due to a significant contribution of the hydrophobic effect.

For better visualization, the phase-solubility diagrams used hereinafter for the stability constant calculations are given in Appendix A. As follows in Appendix A, in acidic buffer solution, the phase solubility diagrams showed an $A_{L}$-type profile, indicating the formation of typical water-soluble complexes. Some anomalous phase-solubility profiles are observed in pH 7.4 buffer solution (Appendix A—an essential deviation of the intercept of the linear phase solubility diagram from $S_{2}^{0}$ was shown. Loftsson et al. [34] noticed the intercept should be equal to $S_{2}^{0}$ and all points should fall on the line drawn at the zero value. Moreover, the authors stated that this situation is often characteristic for the lipophilic poor soluble drugs. The same regularity was shown by us for the structure analogue L-173 [19]. As opposed, for fluconazole a typical $A_{L}$-diagram was obtained by Fernández-Ferreiro with co-workers in water [35]. A specific non-linear phase-solubility profile with a negative deviation at low cyclodextrin concentrations designated as $A_{L}^{-}$ [34] observed in the present study in buffer solution at pH 7.4 is a result of a high lipophilicity of the compounds (ClogP = 2.80 calculated using the program pDISOL-X). Processing of the phase solubility diagrams (pH 7.4) by a linear regression analysis produces a negative intercept, resulting in the theoretically impossible negative values of the stability constants. At the same time, the calculation of the stability constant by a linear regression using $S_{2}^{0}$ leads to its overestimation. As was reported by Loftsson et al. [36], possible reasons for a negative deviation from linearity include, for example, self-association of the drug molecules, drug/cyclodextrin complexes, and non-inclusion complexation. As was shown by Okimoto et al. [37] for the cinnarizine/cyclodextrin complex, a sharp $A_{L}^{-}$-type phase-solubility profile should not be mistaken for an $A_{P}$-type one.

### 4.3. Stability of R-98/RS-186 Complexes with HP-β-Cyclodextrin in pH 2.0 Buffer Solution

The slope values of the $A_{L}$-type phase-solubility diagrams for the studied compounds with 2-HP-β-*CD* in buffer solution at pH 2.0 are lower than unity, indicating the 1:1 stoichiometry of the complexes in the investigated range of cyclodextrin concentrations [38]. The values of the apparent stability constants at different temperatures were calculated by Equation (3) and are shown in Table 2.

According to the classification from the point of view of the optimal bioavailability of hydrophobic drugs stated by Szejtli [39] ($K_{C}^{S}$ from 200 to 5000 M^−1^), the stability constants belong to this range ensuring an optimal drug release and permeation through the lipid layer of the gastrointestinal membranes by the free drug in the case of pharmaceutical formulations. As is shown in Table 2, in acidic solution the stability constants for the complexes of R-98/RS-186 with *CD* decrease with the temperature growth, indicating the exothermic nature of the complexation process. It is a typical situation detected for many drug/*CD* systems [32,40,41] when increased temperature limits the entrapment of the drug molecule in the *CD* cavity. Since a decrease in the stability of the complexes with the temperature growth is observed, an increase of chiral recognition can be proposed at elevated temperatures according to Rekharsky and Innoue [42], who argued that stronger binding between the guest and the host results in a loss of chiral recognition. Interestingly, that replacement of >C=O group (in L-173 molecule—Appendix A) by -CH_2_-CH-OH fragment (R-98/RS-186) results in an increase of the complex stability in acidic buffer at approximately 2.0/2.8-fold (298.15 K) and 1.4/1.8-fold (315.15 K) as compared to L-173. Importantly, the ability for complex formation with *CD* was estimated to be higher in the case of racemic compound; 1.3-fold, as compared to R-isomer (Table 2). These values indicate that 2-HP-β-*CD* forms inclusion complexes more preferentially with RS-186 than R-98, indicating a stereospecific behavior in the presence of *CD*.

### 4.4. Complexation Efficiency of 2-HP-β-CD towards R-98/RS-186, Effect of pH

According to Section 4.2, it was impossible to calculate the stability constants for the compounds in buffer solution at pH 7.4 due to $A_{L}^{-}$ phase-solubility profiles. As was shown by Brewster and Loftsson [43], the complexation efficiency (CE) is a more accurate parameter for the evaluation of the solubilizing effect of cyclodextrins towards the compounds demonstrating an essential deviation of the intercept of the linear phase solubility diagram from $S_{2}^{0}$ (a case when $S_{2}^{0}$ < 0.1 mg/mL). CE is equal to the molar ratio between the complex and free *CD*s concentrations. Table 2 demonstrates the CE values of 2-HP-β-*CD* towards the solubility of R-98/RS-186 calculated by Equation (4) from the slopes of the solubility diagrams. In addition, drug:*CD* molar ratio in solution also correlating to the increase in solubility by *CD* was calculated using Equation (5) (see Table 2). It is evident from the table that according to the CE values, an essentially more pronounced effect of 2-HP-β-*CD* towards the solubility of R-98/RS-186 in the acidic medium (as compared to buffer pH 7.4) is observed. Interestingly, the difference between the CE at different pH is ≈3.4-fold at 298.15 K and ≈67-fold at 315.15 K. Moreover, in acidic medium, a slight increase of the CE with the temperature growth was detected. In opposition, at pH 7.4, a 14- and 17-fold CE decrease was shown for R-98 and RS-186, respectively (Table 2). Comparing the racemic compound and pure enantiomer, 2-HP-β-*CD* causes only a slightly different solubilizing effect on the solubility and at pH 2.0 this effect is leveled with the temperature growth. The data in Table 2 demonstrate that in an acidic medium, the number of molecules in solution which form water soluble complexes reduces from 4 to 3 for both compounds. In turn, at pH 7.4 this number increases from 11 to 145 and from 10 to 171 for R-98 and RS-186, respectively.

### 4.5. Thermodynamics of Complexation, Solubilization and Inherent Dissolution Processes for Chiral Discrimination

The binding equilibrium upon the complex formation is determined by the stabilization of all complexation sources and their mutual interplay, including hydrophobic interactions with the *CD* cavity and van der Waals forces, release of ‘‘high-enthalpy water’’ molecules from the cavity to the bulk solution, hydrogen bonding between the hydroxyl groups of the external surface of *CD* and the drug, and/or electrostatic interactions [44]. In order to disclose comprehensively the impacts of the above-mentioned interactions, the thermodynamic parameters and the driving forces of the complexation and solubilization [19,45] occurring simultaneously in the presence of cyclodextrins were determined. In addition, for the sake of comparison, the thermodynamic functions of inherent dissolution (without *CD*) were calculated and taken into consideration. In the present work, we made attempts to apply the thermodynamic considerations in order to gain more insight in chiral discrimination on the example of enantiomer R-98 and racemic compound RS-186 in acidic buffer pH 2.0. To this end, the respective thermodynamic parameters (changes of free Gibbs energy, enthalpy and entropy) for both samples were calculated (Equations (8)–(15), Section 3) based on the mole fraction unitary units for the standard state and listed in Table 3 for the standard temperature 298.15 K.

It is well known that the change of the Gibbs free energy provides information on the spontaneous/non-spontaneous character of a specific process. The positive values of the Gibbs free energy of inherent dissolution ($\Delta G_{inh/sol}^{0,X}$) show an unfavorable process, as opposed to those of solubilization ($\Delta G_{slbz}^{0,X}$) and complexation ($\Delta G_{C}^{0,X}$) (Table 3) for which the negative values can be attributed to the spontaneous thermodynamically favored solubilization of the compounds in *CD* solutions and inclusion in the hydrophobic cavity of *CD*. A slightly larger negative value for the racemic compound is in agreement with a higher $K_{C}^{X}$ value. As follows from Appendix A, a somewhat more favorable solubilization for the racemic compound as compared to enantiomer takes place upon the increase of *CD* concentration and temperature since increasing concentration promotes the formation of diastereomeric complexes and enhances the enantiomeric discrimination.

Linear plots of $\ln X_{2}$ and $\ln K_{C}^{X}$ versus 1/T applied for the calculation of the dissolution and complexation enthalpies and entropies via Equations (9) and (15) for R98/RS-186 are depicted in Appendix A, respectively. As follows from Table 3, the complex formation for both R-isomer and racemate is driven by favorable negative enthalpy and favorable positive entropy. The observed combination of the thermodynamic functions testifies on behalf of a partial destruction of the solvation shells of the reagents and the dehydration of the cavity by a release of the enthalpy rich water molecules from the *CD* cavity into the bulk solvent leading to the solvent reorganization. A case when different signs of $\Delta H_{C}$ and $T\Delta S_{C}$ are observed can be attributed to a ‘nonclassical’ model of hydrophobic interaction [46,47] approving the impact of the solvation shells reorganization and the hydrophobic effects during the complexation reaction. Negative enthalpy values resulting in a favorable drop in enthalpy arise from the displacement of water molecules from the cavity by suitable guest molecules with less polarity than water [48] and are typical for low energy interactions [45]. In turn, relatively large positive $\Delta S_{C}$ values are due to the disordering of the water layers around the complex which are less ordered or contain a smaller number of water molecules with respect to free reactants. In addition, negative values of the enthalpy suggest the impact of the dipole interactions and/or hydrogen bonding to the complexation process. A more negative enthalpy for the racemic compound can be explained by a more readily formation of hydrogen bonds upon the complexation in the presence of another enantiomer in solution [49]. The absolute values of the enthalpy and entropy terms are comparable and are very close for R-98, at the same time, a ratio ǀ$\Delta H_{C}^{0,X}$ǀ > ǀ$T\Delta S_{C}^{0,X}$ǀ at about 1.5-fold for RS-186 once more proved a role of the interactions different than hydrophobic forces in the complexation process [50].

As opposed to the complexation, the process of inherent dissolution is unfavorable according to the positive changes of $\Delta H_{inh/sol}^{0,X}$ and negative—$T\Delta S_{inh/sol}^{0,X}$. Obviously, it is explained by “squeezing” of the hydrophobic compound’s molecules from the water structure and decreasing the amount of the dissolved substance. At the same time, the *CD* molecules which include the hydrophobic cavities and external hydrophilic -OH groups provoke interactions both with the cavity and with surface, thus increasing the solubility. Interestingly, the solubilization process is characterized as unfavorable from positive $\Delta H_{slbz}^{0,X}$ (like inherent dissolution) and favorable from also positive $T\Delta S_{slbz}^{0,X}$ (like complexation). At the same time, according to inequality $\Delta H_{slbz}^{0,X}$ < $T\Delta S_{slbz}^{0,X}$, the solubilization is governed by a favorable entropy rather than enthalpy similarly to the complexation process. To the aim of quantitatively evaluation of the chiral discrimination of enantiomer and racemic compound, it seems very useful to calculate all the excess thermodynamic parameters using Equations (17)–(19). The derived values are given in Table 3. For both compounds the Gibbs free energy of the complexation process is more negative as compared to the solubilization indicating a larger driving force of the complexation to the solubility enhancement. At the same time, a rather small difference between the driving force of the two investigated compounds ($\Delta \Delta G_{C}^{0,X}$ = $\Delta \Delta G_{slbz}^{0,X}$ = −0.7) is shown. Notably, since an inequality |$\Delta \Delta H^{0,X}$| > |$T\Delta \Delta S^{0,X}$| realizes both for the complexation and solubilization phenomena, this difference arises mostly from the enthalpy gain. Similarly, the excess $\Delta \Delta G_{inh/sol}^{0,X}$ parameter also comes in more extent from the positive enthalpy term. From the thermodynamic point of view, besides the involvement of the hydrophobic effects to the complexation process, attractive van der Waals interactions can be proposed and are shown to be larger for the racemic compound (more negative $\Delta H_{C}^{0,X}$ and less positive $T\Delta S_{C}^{0,X}$). As follows, RS-186 molecules are strongly bounded within the *CD* cavity than those of R-98 (in fully compliance with inequality $K_{C}^{X}$ (R-98) < $K_{C}^{X}$ (RS-186)), highly probable due to a loss of some freedom of motion of RS-186 as compared to R-98.

At the next step of the investigation, it seemed useful to trace the impact of the hydrophobic interactions on the chiral discrimination upon the complex formation. To this end, an approach described in Section 3 (Issue 1) [20] was applied and the correlations between the changes of the free energies of the complexation ($\Delta G_{C}^{T,X}$) and inherent solubility ($\Delta G_{sol}^{T,X}$) were derived using mole fraction units and plotted (Figure 3).

A linear correlation between $\Delta G_{C}^{T,X}$ and $\Delta G_{sol}^{T,X}$ via Equation (16) results in the following slope and intercept: R-98: a = −0.776, b = 7.131, r = 0.998; RS-186: a = −0.611, b = 2.125, r = 0.980. According to the literature [20], a negative slope of a linear plot (Figure 3) evidences the contribution of the hydrophobic interactions at about 78%, and 61% of the driving force for complex stability for R-98 and RS-186, respectively. This result is in agreement with the conclusions from the complexation thermodynamic parameters, when a more negative $\Delta H_{C}^{0,X}$ for RS-186, and a more positive $\Delta S_{C}^{0,X}$ for R-98 were estimated (Table 3) and testifies to chiral recognition.

### 4.6. Enantioselective Membrane Permeability through the PermeaPad Barrier

The membrane assisted separation of enantiomers is a method often used in chiral discrimination. This fact suggests possible differentiation of the permeation behavior of drugs and physiologically active substances through the biological barriers. On the other hand, in the case of cyclodextrin based drug formulations the permeation process can be complicated due to a reduction of the amount of free drug molecules and growing the viscosity of the stock solution [51]. In the present study, the permeability coefficients ($P_{app}$) for both R-98 and RS-186 were determined through the biomimetic PermeaPad barrier in pure buffers (pH 2.0 and pH 7.4) and in the presence of two *CD* concentrations in donor solution in order to reveal the influence of *CD* on the permeability of both racemic compound and enantiomer. The initial experimental data used for the $P_{app}$ calculation and the resultant cumulative amounts of the permeated drug, steady state fluxes and permeability coefficients are given in Table 4.

It seemed that higher permeability coefficients in pH 7.4 as compared to pH 2.0 (charged species) would be expected. However, as follows from the presented results, the permeability coefficients of both compounds in both pure buffers demonstrated very close $P_{app}$ values. At the same time, very close permeability coefficients were also estimated in the same buffers in our more recent studies for fluconazole (FCZ) [52] and novel thiazolidine-2,4-dione derivative [53]. Interestingly, $P_{app}$ (R/RS) < $P_{app}$ (FCZ) at approximately 2.1-fold, most probably due to a difference of the molecular mass, and, as follows, the diffusion coefficients of the compounds [54].

Both in acidic medium and at pH 7.4 the permeability, there follows an expected regularity opposed to the solubility: $P_{app}$ (R-98) < $P_{app}$ (RS-186) and $S_{2}$ (R-98) > $S_{2}$ (RS-186) thus demonstrating a stereoselective and excretive transport. At the same time, only for R-98 the permeability in pH 7.4 where the neutral species of the compound exists exceeds the permeation rate of charged particles in buffer pH 2.0. Unexpectedly, non-typical situation when $P_{app}$ (RS-186) in acidic solution is slightly higher than in pH 7.4 can possibly be attributed to a less conformational freedom of RS-compound as compared to pure R-enantiomer [45] which disturbs a proper membrane penetration process. A comparative analysis of the influence of *CD* on the permeation of R-enantiomer and RS-compound (Table 4) revealed that a 9.15- and 8.72-fold permeability reduction occurs up to 0.03 M concentration of *CD* in donor solution at pH 2.0 for R-98 and RS-186, respectively. Notably, the influence of *CD* on the permeability decreases in buffer pH 7.4 is practically the same: a 9.66- and 8.89-fold, respectively. According to the approach proposed in [51] the plots illustrating a relationship between the solubility and permeability upon the addition of *CD* concentrations at both investigated pHs were constructed and depicted in Figure 4. It is well known that the permeability varies over a more narrow range than the solubility does [55] and, expectedly, the permeability decrease can be of orders of magnitude smaller upon a great solubility growth. As it is evident from Figure 4, in acidic medium at 0.01 M *CD* concentration $P_{app}$ of R-98 is 2.2-fold higher than for RS-186. In turn, at pH 7.4 this difference is only 1.1-fold. Interestingly, that at 0.03 M *CD* this difference is smoothed out.

Notably, a dramatic decrease of the permeability coefficient up to 0.01 M *CD* concentration was observed in both studied media, whereas from 0.01 to 0.03 M *CD* concentration the permeability practically does not change. The same regularity was shown in our previous studies [52] and was explained by an abrupt decrease of the free drug amount even at relatively low *CD* concentrations. At the same time, taking into account the lipophilic nature of the phospholipid membrane, the PermeaPad barrier, such a property as lipophilicity can also contribute to the permeability behavior. According to the classification estimated by the developers of PermeaPad [18], both R-98 and RS-186 can be attributed to medium permeable substances. At the same time, even in the presence of 0.01 M *CD* their permeability is classified as low. It was interesting to make a raw approximation of cell permeability of the synthesized compounds based on the comparison with the well-known fluconazole. On the one hand, a 2.1-fold greater $P_{app}$ of FCZ as compared to R/RS-compounds was indicated above. On the other one, the FCZ Caco-2 monolayers permeability from the literary review was shown to be $P_{Caco-2}$ = 1.16 × 10^−5^ cm·s^−1^ [56], $P_{Caco-2}$ = 2.95 × 10^−5^ cm·s^−1^ [57], and $P_{app}$ (PermeaPad) = 1.9 × 10^−5^ cm·s^−1^ [52]. As follows, proportionally, the permeability from the Caco-2 model is in close consequence (taking into account some discrepancies in the literature issues) with the results from the PermeaPad and allows to propose $P_{Caco-2}$ of the synthesized substances at around (9.2 ± 0.9) × 10^−6^ cm·s^−1^, which corresponds to medium permeability by the classification of Manning et al. [58].

## 5. Conclusions

Novel potential antifungal compound of 1,2,4-triazole class have been synthesized and characterized as pure enantiomer (R-98) and racemic compound (RS-186). Phase solubility studies in 2-HP-β-*CD* solutions (298.15–313.15 K) showed $A_{L}$- and $A_{L}^{-}$-type phase-solubility profiles in buffer solutions at pH 2.0 and pH 7.4, respectively. The racemic compound formed more stable complexes ($K_{C}^{S}$ = 708.9 M^−1^) with *CD* as compared to R-isomer ($K_{C}^{S}$ = 516.8 M^−1^). An increase of chiral recognition at elevated temperatures was stated. The chiral discrimination based on complex consideration of the complexation/solubilization/inherent dissolution thermodynamic parameters, including the excess differentiation parameters between the racemic compound and R-enantiomer revealed the following issues: (1) unfavorable inherent dissolution and thermodynamically favored solubilization and complexation (more favorable for racemic) due to the inclusion of compound in the hydrophobic cavity of *CD*; (2) a larger influence of the complexation as compared to solubilization on the driving force of the solubility enhancement; (3) a 1.3-fold more pronounced impact of the hydrophobic interactions to the driving force of the complex formation for R-enantiomer, and a larger contribution of the attractive van der Waals interactions for the racemic compound; (4) stronger binding of racemic than of R-isomer within the *CD* cavity due to a loss of some freedom of motion for racemic compound. In addition, a 2.2-fold higher permeability coefficient through the lipophilic membrane, PermeaPad barrier of R-98 as compared to RS-186 in acidic solution at 0.01 M *CD* was estimated. Accordingly, chiral discrimination using 2-HP-β-*CD* as a chiral selector was shown to be appropriate for the investigated objects.

We hope that the findings of this investigation will have an impact on the chiral recognition and may be useful in designing pharmaceutical formulations based on the studied compounds.

## Figures and Tables

**Figure 1 pharmaceutics-14-00864-f001:**
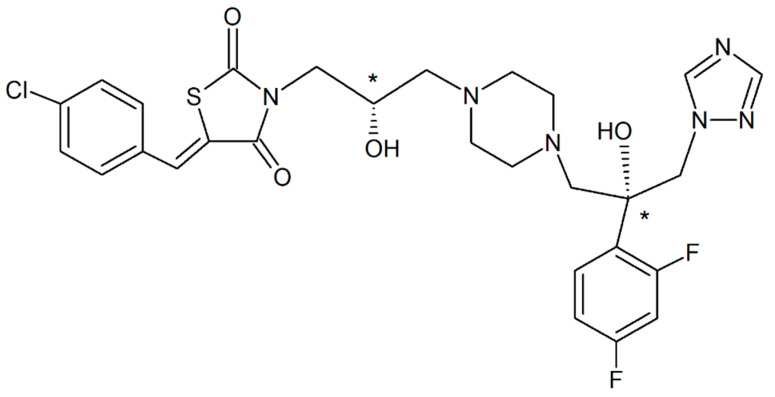
Structure of the studied compounds. Chiral centers are marked by *.

**Figure 2 pharmaceutics-14-00864-f002:**
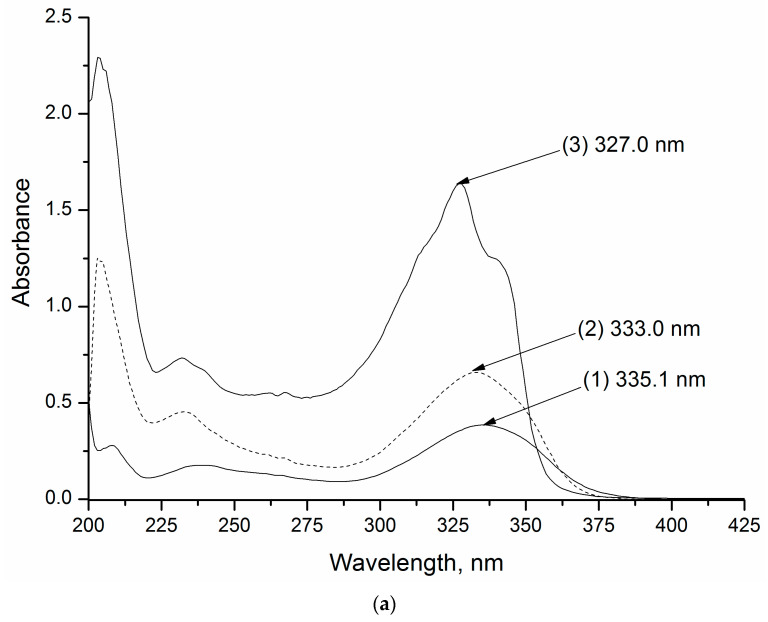
The absorption spectra of RS-186 in pure buffer (1), with 0.05 M 2-HP-β-*CD* (2) and n-hexane (3); pH 2.0—(**a**), pH 7.4—(**b**). Solid lines mean the spectra of pure solvents; dashed lines indicate the spectra of aqueous 2-HP-β-*CD* solutions.

**Figure 3 pharmaceutics-14-00864-f003:**
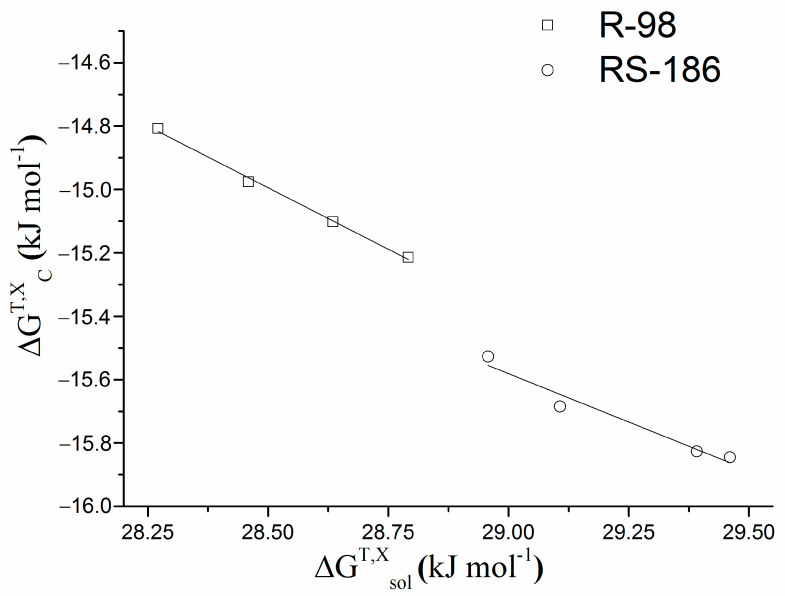
Correlation between the free energy of complex formation ($\Delta G_{C}^{T,X}$) and free energy of the inherent solubility ($\Delta G_{sol}^{T,X}$) for R-98 (black opened squares) and RS-186 (black opened circles) at pH 2.0 (mole fraction scale).

**Figure 4 pharmaceutics-14-00864-f004:**
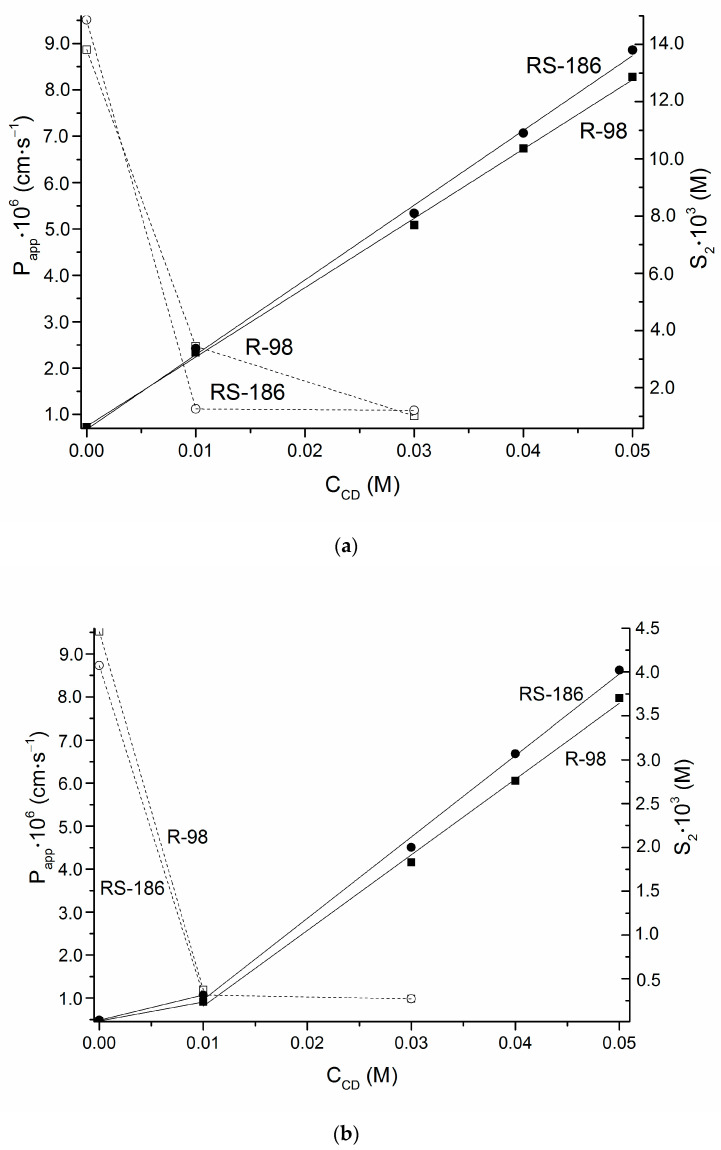
Solubility-permeability interrelation exemplified by the effect of *CD* concentration on R-98: apparent permeability coefficient—black opened squares—□, solubility—black filled squares—■; and R-186: permeability coefficient—black opened circles—○, solubility—black filled circles—●. (**a**) pH 2.0; (**b**) pH 7.4. The solid and dashed lines indicate the solubility and permeability dependences, respectively.

**Table 1 pharmaceutics-14-00864-t001:** Experimental solubility ($S_{2}$) in buffer pH 2.0 and buffer pH 7.4 with different 2-HP-β-*CD* concentrations. ($C_{CD}$) in the temperature range of 298.15–313.15 K and pressure *p* = 100 kPa.

$C_{CD}/$ M	pH 2.0				pH 7.4			
	298.15 K	303.15 K	310.15 K	313.15 K	298.15 K	303.15 K	310.15 K	313.15 K
	$S_{2}\;10^{3}/M$				$S_{2}\;10^{4}/M$			
	R-98							
0	0.62 ± 0.01	0.69 ± 0.02	0.83 ± 0.03	0.87 ± 0.03	0.18 ± 0.01	0.21 ± 0.01	0.25 ± 0.01	0.27 ± 0.01
0.01	3.23 ± 0.10	3.59 ± 0.11	4.12 ± 0.12	4.20 ± 0.11	2.35 ± 0.03	1.79 ± 0.04	1.42 ± 0.04	0.58 ± 0.02
0.03	7.68 ± 0.11	8.21 ± 0.21	8.86 ± 0.23	9.49 ± 0.25	18.30 ± 0.41	9.65 ± 0.28	4.43 ± 0.12	1.86 ± 0.04
0.04	10.36 ± 0.23	10.84 ± 0.30	12.50 ± 0.39	12.69 ± 0.41	27.60 ± 0.55	13.90 ± 0.41	6.07 ± 0.18	2.62 ± 0.07
0.05	12.86 ± 0.22	14.04 ± 0.41	15.72 ± 0.38	16.15 ± 0.44	37.00 ± 0.52	18.60 ± 0.33	8.08 ± 0.21	3.33 ± 0.08
	RS-186							
0	0.50 ± 0.01	0.54 ± 0.01	0.61 ± 0.02	0.69 ± 0.02	0.27 ± 0.01	0.30 ± 0.01	0.35 ± 0.01	0.37 ± 0.01
0.01	3.37 ± 0.08	3.40 ± 0.04	3.64 ± 0.04	3.97 ± 0.03	3.15 ± 0.02	2.74 ± 0.03	1.93 ± 0.03	1.37 ± 0.06
0.03	8.10 ± 0.22	8.37 ± 0.18	8.68 ± 0.19	9.30 ± 0.31	20.00 ± 0.32	10.60 ± 0.33	5.01 ± 0.05	2.50 ± 0.03
0.04	10.90 ± 0.31	11.20 ± 0.19	11.79 ± 0.33	12.90 ± 0.33	30.70 ± 0.51	15.60 ± 0.41	6.97 ± 0.08	3.20 ± 0.05
0.05	13.80 ± 0.40	14.20 ± 0.22	14.88 ± 0.24	15.90 ± 0.40	40.20 ± 0.58	20.10 ± 0.21	8.58 ± 0.08	3.65 ± 0.05

Each solubility value represents the mean ± SD (*n* ≥ 3). The standard uncertainties are *u*(*T*) = 0.15 K, *u*(*p*) = 3 kPa.

**Table 2 pharmaceutics-14-00864-t002:** Complexation efficiency (CE), apparent stability constant ($K_{C}^{S}$) and Molar Ratio (compound:2-HP-β-*CD*) at different temperatures and pressure *p* = 100 kPa.

*T*/K	CE/%		$K_{C}^{S}/M^{-1}$		Molar Ratio (Drug:*CD*)	
	pH 2.0	pH 7.4	pH 2.0	pH 7.4	pH 2.0	pH 7.4
	R-98					
298.15	32.0	9.5	516.8 ± 51.1	n.d. *	1:4	1:11
303.15	35.5	4.4	506.6 ± 49.3	n.d.	1:4	1:24
310.15	40.7	1.7	493.6 ± 49.0	n.d.	1:3	1:60
313.15	42.7	0.7	486.9 ± 48.8	n.d.	1:3	1:145
	RS-186					
298.15	35.4	10.3	708.9 ± 71.4	n.d.	1:4	1:10
303.15	36.9	4.6	688.0 ± 60.3	n.d.	1:4	1:23
310.15	39.1	1.7	646.7 ± 61.5	n.d.	1:3	1:59
313.15	43.2	0.6	623.6 ± 60.0	n.d.	1:3	1:171

* n.d.—not determined due to $A_{L}^{-}$-type phase-solubility diagram.

**Table 3 pharmaceutics-14-00864-t003:** Complexation constant ($K_{C}^{X}$), thermodynamic parameters of inclusion complex formation of the studied compounds with cyclodextrins: change of the standard Gibbs energy ($\Delta G_{C}^{0,X}$), enthalpy ($\Delta H_{C}^{0,X}$), and entropy ($T\Delta S_{C}^{0,X}$); solubilization thermodynamic functions: change of the standard Gibbs energy ($\Delta G_{slbz}^{0,X}$), enthalpy ($\Delta H_{slbz}^{0,X}$), and entropy ($T\Delta S_{slbz}^{0,X}$); and inherent solubility thermodynamic parameters: in buffer solution pH 2.0 at 298.15 K and pressure *p* = 100 kPa (mole fraction scale).

$K_{C}^{X}$	$\Delta G_{C}^{0,X}/$ kJ·mol^−1^	$\Delta H_{C}^{0,X}/$ kJ·mol^−1^	$T\Delta S_{C}^{0,X}/$ kJ·mol^−1^	$\Delta {G_{slbz}^{0,X}}^{a}/$ kJ·mol^−1^	$\Delta H_{slbz}^{0,X}/$ kJ·mol^−1^	$T\Delta S_{slbz}^{0,X}/$ kJ·mol^−1^	$\Delta G_{inh/sol}^{0,X}/$ kJ·mol^−1^	$\Delta H_{inh/sol}^{0,X}/$ kJ·mol^−1^	$T\Delta S_{inh/sol}^{0,X}/$ kJ·mol^−1^
R-98									
392.9 ± 38.8	−14.8	−7.2 ± 0.8	7.6 ± 0.9	−7.5	11.8 ± 0.1	19.3 ± 0.5	28.2	18.5 ± 0.8	−9.7 ± 0.6
RS-186									
525.2 ± 52.9	−15.5	−9.3 ± 1.3	6.2 ± 0.9	−8.2	6.8 ± 1.5	15.0 ± 3.3	30.0	19.0 ± 2.2	−11.0 ± 1.5
	Excess thermodynamic parameters								
	$\Delta \Delta G_{C}^{0,X}$ ^b^	$\Delta \Delta H_{C}^{0,X}$ ^e^	$T\Delta \Delta S_{C}^{0,X}$ ^e^	$\Delta \Delta G_{slbz}^{0,X}$ ^c^	$\Delta \Delta H_{slbz}^{0,X}$ ^e^	$T\Delta \Delta S_{slbz}^{0,X}$ ^e^	$\Delta \Delta G_{inh/sol}^{0,X}$ ^d^	$\Delta \Delta H_{inh/sol}^{0,X}$ ^e^	$T\Delta \Delta S_{inh/sol}^{0,X}$ ^e^
	−0.7	−2.1	−1.4	−0.7	−5.0	−4.3	1.8	0.5	−1.3

^a^ calculated at maximal 0.05 M *CD* concentration; ^b^ Equation (17); ^c^ Equation (18); ^d^ Equation (19); ^e^ $\Delta \Delta H^{0,X}$ and $T\Delta \Delta S^{0,X}$ calculated by the difference between the respective parameters of racemic and enantiomer (see Section 3, Issue 2).

**Table 4 pharmaceutics-14-00864-t004:** Donor solution concentrations, cumulative amounts per unit area permeated in 5 h, steady state fluxes (J), and permeability coefficients ($P_{app}$) of R-98 and RS-186 in the presence of *CD* in buffer solutions (pH 2.0 and pH 7.4).

*CD* Concentration/M	Donor Solution Concentration/M	Cumulative Amount Permeated/µM·cm^−2^	Steady State Flux (J)/ µM·cm^−2^·s^−1^	$\mathrm{Permeability}\;\mathrm{Coefficient}\;(P_{app})/\mathrm{cm}\cdot s^{-1}$
pH 2.0				
R-98				
0	2.89 × 10^−^^4^	3.58 × 10^−^^8^	2.56 × 10^−^^6^	(8.87 ± 0.42) × 10^−^^6^
0.01087	5.89 × 10^−^^4^	6.67 × 10^−^^8^	1.46 × 10^−6^	(2.47 ± 0.19) × 10^−6^
0.02899	7.42 × 10^−^^4^	4.10 × 10^−8^	7.19 × 10^−^^7^	(9.69 ± 0.29) × 10^−^^7^
RS-186				
0	2.51 × 10^−^^4^	4.12 × 10^−^^8^	2.39 × 10^−^^6^	(9.51 ± 0.48) × 10^−^^6^
0.01087	6.60 × 10^−^^4^	2.51 × 10^−^^8^	7.39 × 10^−^^7^	(1.12 ± 0.09) × 10^−6^
0.02899	1.41 × 10^−^^3^	1.12 × 10^−^^7^	1.54 × 10^−^^6^	(1.09 ± 0.11) × 10^−^^6^
pH 7.4				
R-98				
0	2.28 × 10^−^^5^	4.46 × 10^−^^9^	2.17 × 10^−^^7^	(9.52 ± 0.63) × 10^−^^6^
0.01087	3.00 × 10^−^^4^	6.39 × 10^−^^9^	3.58 × 10^−^^7^	(1.19 ± 0.06) × 10^−6^
0.02899	7.05 × 10^−4^	3.58 × 10^−^^8^	6.95 × 10^−^^7^	(9.85 ± 0.30) × 10^−^^7^
RS-186				
0	2.82 × 10^−^^5^	4.46 × 10^−^^9^	2.46 × 10^−^^7^	(8.73 ± 0.68) × 10^−^^6^
0.01087	3.81 × 10^−^^4^	6.72 × 10^−^^9^	3.08 × 10^−^^7^	(1.07 ± 0.06) × 10^−^^7^
0.02899	2.90 × 10^−^^4^	6.19 × 10^−^^9^	2.84 × 10^−^^7^	(9.82 ± 0.49) × 10^−^^7^

## Data Availability

The results obtained for all experiments performed are shown in the manuscript and SI, the raw data will be provided upon request.

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
