# Peer review of "Chiral Recognition R- and RS- of New Antifungal: Complexation/Solubilization/Dissolution Thermodynamics and Permeability Assay"

_pharmaceutics, 2022, doi:10.3390/pharmaceutics14040864_

Round 1

Reviewer 1 Report

08.04.2022.

A review to evaluate its suitability for publication Type of manuscript:

Article
Title: Chiral recognition R- and RS- of new antifungal: Complexation/solubilization/dissolution thermodynamics and permeability assay

Authors: Tatyana V. Volkova, Olga R. Simonova, Igor B. Levshin, German L. Perlovich

The manuscript by German L. Perlovich and co-workers described the identification of conditions governing processes in chiral recognition solutions.

The materials of the manuscript submitted for review are of significant interest to pharmacy since they are aimed at solving such important tasks as:

  1. Development of a new medicinal substance
  2. Development of enantioselective synthesis and search for an enantioselective reagent.
  3. А promising solution to the issue of antibiotic resistance.

The statement of the problem is clearly correlated with the results.

However, there are several questions, the answers to which will significantly improve the quality of the submitted manuscript material:

  1. Lines 65-67:

The structure of the newly synthesized triazole derivative is a cross between the structure of ketoconazole (imidazole) and fluconazole. Is it possible to describe its molecule by the name of IUPAC?

An important question concerning the centers of chirality of the molecule and the subsequent mechanisms of interaction with the enantioselective reagent:

  1. How many asymmetric carbon atoms?
  2. Is that the carbon in the ethanol residue?
  3. There are no results of the polarimetry method in the work - what is the value of the specific rotation for the new triazole compound?

  1. Lines 95-97:

What is the degree of purity (or an API content) in the RS-186 and R-98 components? This is important because impurities in the form of optically active substances can affect the mechanism of the processes studied in this work?

  1. Where is the Figure S1 (a) and (b) of Synthetic procedure?
  2. Line 573: What is the value of Predicted or Experimental log P Properties for RS-186 and R-98 components? Are differences between them?

Respectfully, reviewer

Author Response

Reply to Reviewer_1:

The manuscript by German L. Perlovich and co-workers described the identification of conditions governing processes in chiral recognition solutions.

The materials of the manuscript submitted for review are of significant interest to pharmacy since they are aimed at solving such important tasks as:

  1. Development of a new medicinal substance
  2. Development of enantioselective synthesis and search for an enantioselective reagent.
  3. А promising solution to the issue of antibiotic resistance.

The statement of the problem is clearly correlated with the results.

However, there are several questions, the answers to which will significantly improve the quality of the submitted manuscript material:

Comment:

Lines 65-67:

  1. The structure of the newly synthesized triazole derivative is a cross between the structure of ketoconazole (imidazole) and fluconazole. Is it possible to describe its molecule by the name of IUPAC?

Reply:

The IUPAC name of the molecule has been added to the manuscript.

Comment:

An important question concerning the centers of chirality of the molecule and the subsequent mechanisms of interaction with the enantioselective reagent:

  1. How many asymmetric carbon atoms?

Reply:

Two chiral centers are presented in the studied compound: in the structure of 3-substituted (2-hydroxypropyl)thiazolidine-2,4-dione, and in the triazole structure. These centers are indicated in Figure 1 by asterisk.

Comment:

  1. Is that the carbon in the ethanol residue?

Reply:

Yes, the carbon in the structure of 3-substituted (2-hydroxypropyl)thiazolidine-2,4-dione, and in the triazole structure.

Comment:

  1. There are no results of the polarimetry method in the work - what is the value of the specific rotation for the new triazole compound?

Reply:

The polarimetry experiments (Phenomenex Cell-4 45mm EtOH/AcN 10/90 23 C Detector UV/CD 260 nm 1 ml/min) didn't detect the differences between R- and RS- compounds. The ratio of two peaks in the spectrum equaled to 50:50 indicated the presence of about equal amounts of two geometric enentiomers. We didn't make the discriminations of enantiomers on a chiral column.

Comment:

  1. Lines 95-97:

What is the degree of purity (or an API content) in the RS-186 and R-98 components? This is important because impurities in the form of optically active substances can affect the mechanism of the processes studied in this work?

Reply:

The degree of purity estimated using HPLC (Shimadzu LC-20 AD, System  - FOC Colon- Kromasil -100-5mkm. C18, 4.6x250 mm., N 62511. Elution: A - H3PO4 0.01  pH 2.6; B - MeCN, fl -1.0 ml/min, loop 20 mkl, and Phenomenex column C18, 150*, 3.3 mm, column temperature 30оС (mobile phase: acetonitrile), m/z 619,1810; retention time: 10.4- 10.5 min.) was 96% and 97% for the RS-186 and R-98 components. The purity was additionally approved by 1H NMR, 13C NMR spectra (the information has been added in SI file).

Comment:

  1. Where is the Figure S1 (a) and (b) of Synthetic procedure?

Reply:

Figure S1 (a) and (b) of Synthetic procedure are in the Supporting Information file.

This information has been inserted in section "2.2.1. Synthesis".

Comment:

  1. Line 573: What is the value of Predicted or Experimental log P Properties for RS-186 and R-98 components? Are differences between them?

Reply:

The calculated value ClogP=2.80 is shown in section "4.2. Solubility studies". Unfortunately, the discrimination of the studied compounds by the calculated (from the structure) logP is difficult. Determination of the logP parameter for RS-186 and R-98 experimentally is a subject of our future scientific research.

Reviewer 2 Report

The article entitled “Chiral recognition R- and RS- of new antifungal: Complexation/solubilization/dissolution thermodynamics and permeability assay” deals with the investigation of β-cyclodextrin and 2-hydroxypropyl-β-cyclodextrin as suitable chiral selector for R- and RS- of novel synthetized antifungal drug. The article is well-written and structured. The topic is interesting and innovative as chiral selection is an essential tool in drug discovery. I have only few suggestions:

Please add the degree of substitution in case of 2-hydroxypropyl-β-cyclodextrin to the Materials section!

What was the sensitivity of UV spectroscopic method? Please add LOD and LOQ to the method description!

The authors mention in line 340 “solubility experiments in β-cyclodextrin solutions were tried but appeared unsuccessful, since an insoluble complex precipitated”. Please explain what can be the reason of precipitation!

Author Response

Reply to Reviewer_2:

The article entitled “Chiral recognition R- and RS- of new antifungal: Complexation/solubilization/dissolution thermodynamics and permeability assay” deals with the investigation of β-cyclodextrin and 2-hydroxypropyl-β-cyclodextrin as suitable chiral selector for R- and RS- of novel synthetized antifungal drug. The article is well-written and structured. The topic is interesting and innovative as chiral selection is an essential tool in drug discovery. I have only few suggestions:

Comment:

Please add the degree of substitution in case of 2-hydroxypropyl-β-cyclodextrin to the Materials section!

Reply:

The degree of substitution (0.6 molar substitution) has been added to the Materials section.

Comment:

What was the sensitivity of UV spectroscopic method? Please add LOD and LOQ to the method description!

Reply:

The LOD and LOQ parameters characterizing the sensitivity of UV spectroscopic method (calculated according to the literature guidelines [International Conference on Harmonization of Technical Requirements for Registration of Pharmaceuticals for Human Use, ICH Harmonisation Tripartite Guideline. Validation of Analytical Procedures: Text and Methodology Q2 (R1), Complementary Guideline on Methodology, London, UK, November 1996.]) have been added to the method description.

Comment:

The authors mention in line 340 “solubility experiments in β-cyclodextrin solutions were tried but appeared unsuccessful, since an insoluble complex precipitated”. Please explain what can be the reason of precipitation!

Reply:

The precipitation of the insoluble complex of highly lipophilic compounds with β-cyclodextrin (B-type solubility diagram) in most cases is a consequence of β-cyclodextrin limited aqueous solubility (about 18.5 mg·mL-1), and formation of poorly soluble inclusion complexes [Del Valle, Process Biochemistry 39 (2004) 1033–1046; Loftsson et al. Int. J. Pharm. 302 (2005) 18–28].

The respective explanation has been added to section 4.2. Solubility studies.
